# Elevated congenital heart disease birth prevalence rates found in Central Vietnam and dioxin TCDD residuals from the use of *2, 4, 5-T herbicides* (Agent Orange) in the Da Nang region

Hoang Thi Nam Giang[1]*, Tran Thanh Hai[2], Hoang Nguyen[3], Thanh K. Vuong[4], Lois Wright Morton[5], Casey B. Culbertson[6]

1 School of Medicine and Pharmacy, The University of Da Nang, Da Nang, Vietnam, 2 Da Nang Hospital for Women and Children, Da Nang, Vietnam, 3 Department of Pediatric Cardiology, UT Southwestern Medical Center, Dallas, Texas, United States of America, 4 City Children's Hospital, Ho Chi Minh City, Vietnam, 5 College of Agriculture and Life Sciences, Iowa State University, Ames, Iowa, United States of America, 6 MD1World, San Diego, California, United States of America

* htngiang@ud.edu.vn

**Data Availability Statement:** All data are in the manuscript.

## Abstract

Congenital heart disease (CHD) birth prevalence rate in Da Nang City and two adjacent provinces in Central Vietnam reported by Giang et al. in 2019 was 20.09/1000 births, much higher than any CHD birth rates previously reported. In this current study, three physicians trained in pediatric cardiology reanalyzed and reclassified the Giang et al 2019 cardiac anomalies data, eliminating singular small PDAs and separating cardiac defects into 27 contemporary CHD subgroups. These CHD subgroups were then statistically compared with Liu et al. 2019 Global CHD birth prevalence study of Asian Low-Middle Income Countries (LMIC) CHD subgroup rate of 9.34/1000 births (95% CI 8.07–10.70). Despite applying newer diagnostic criteria and refining the cardiac anomalies data, the Da Nang region continued to show significantly (p<0.0001) elevated total CHD birth prevalence rates at 14.71/1000 births (95% CI 12.74–16.69) compared to the Asian LMIC CHD birth prevalence rate 9.34/1000 births. This finding raises the question of whether environmental persistence of the contaminant dioxin TCDD from *2,4,5-T herbicides* (Agent Orange) used during the Vietnam War (1961–1971) in the Da Nang region might be a factor associated with elevated CHD birth prevalence, as it is not present in other LMIC surrounding Vietnam. We recommend testing of soils and sediments in rural and agricultural areas in Central Vietnam that received high volume applications of contaminated herbicides to assess the relationship of the higher CHD birth prevalence rate and the presence of residual dioxin TCDD. Enhanced fetal cardiac echocardiography in the region to screen for CHD would enable early interventions and could improve outcomes for infants and children.

**Funding:** The authors received no specific funding for this work.

**Competing interests:** The authors have declared that no competing interests exist.

## Background

The burden of congenital heart disease (CHD) remains a serious global health problem in low-middle income countries (LMIC). Although there is increasing interest in supporting CHD surgical programs in LMIC ("developing") countries, early diagnosis and timely treatment of CHD is only routinely available to children born in developed countries [1,2]. In 2019, Giang et al. published birth prevalence data on all congenital anomalies diagnosed between 2015–2016 at the Da Nang Hospital for Women and Children's Hospital (DHWC) of which approximately 55% of the births were from pregnancies in Da Nang City and the remaining 45% from pregnancies in the adjacent provinces of Quang Nam and Quang Ngai in Central Vietnam. This single center study showed a total congenital anomalies prevalence rate of 38.44/1000 births, with a CHD birth prevalence rate of 20.09/1000 births (52.3% of all anomalies found). Authors suggest that the TCDD dioxin contaminated herbicide Agent Orange, which was used extensively in the Da Nang region during the Vietnam War (1961–1971), might be a potential source of variation in CHD rates [3]. Concurrent with the original Giang et al. 2019 study, an updated Global birth prevalence of CHD defects systematic review and meta-analysis of 260 studies (1970–2017) was published by Liu et al., which when restricted to Asia (74 studies-52,206,662 live births) reported a CHD birth prevalence rate of 9.34/1000 births (95% CI 8.07–10.70) in LMIC surrounding or nearby to Vietnam [4].

The primary goal of the current study is to conduct a detailed re-evaluation of the original study's cardiac anomalies data to better define the total CHD birth prevalence rate in the Da Nang region as well as to separate those cardiac defects initially found into 27 contemporary CHD subgroups. Wherever possible, we then determine the 27 CHD subgroup prevalence rates. Finally, both the Da Nang total CHD prevalence rate as well as subgroup CHD prevalence rates are compared to the recently published Liu et al. Asian CHD total and subgroup birth prevalence rates to determine if there are statistically significant differences between the two studies CHD birth prevalence rates.

A second goal of the current study is a preliminary exploration of possible association between the high Da Nang region CHD birth prevalence rates and the extensive use of herbicides below the 17th parallel during the Vietnam War. The United States and Republic of Vietnam between 1961 and 1971 sprayed more than 20.2 million gallons (72+ Million Liters) of various herbicides on South Vietnam, including in the Da Nang region. One of the most widely used herbicides, Agent Orange, was a mixture of butoxyethanol esters of *2,4-dichlorophenoxyacetic acid* (2,4-D) and *2,4,5-trichlorophenoxyacetic acid (2,4,5-T)* [5–8] contaminated by the dioxin TCDD (*2,3,7,8-tetracholorodibenzo-p-dioxin*) [6–8]. Fig 1 is a geographical re-creation of Stellman et al. 2003 published data of known records of *2,4,5-T herbicide* applications in the Da Nang region overlaid on the major soil groups of the region. Areas receiving higher volumes of herbicides contaminated with the dioxin TCDD are concentrated northwest to southeast along the interior lowlands and river valleys in Quang Nam and Quang Ngai provinces. Note the color orange denotes areas receiving 4,800 liters or more of herbicides. High levels of herbicide concentrations are likely the result of multiple aerial applications over the same area during the ten-year war period and the storage and use of herbicides at military bases in the region.

There have been widespread assumptions that herbicide formulations using 2,4,5-T with the contaminate dioxin TCDD degrade and are eliminated from the environment by photo degradation, volatilization into the atmosphere, or microbial activities within days to 1–2 years [8–10]. However, not all the dioxin TCDD residues have degraded over the last six decades in seriously contaminated locations in South Vietnam; especially military airbases, base camps, chemical storage locations, dumping and burial areas, and heavily sprayed areas with topographic low spots where runoff and soil erosion may have concentrated dioxin TCDD [11]. As a result of water and

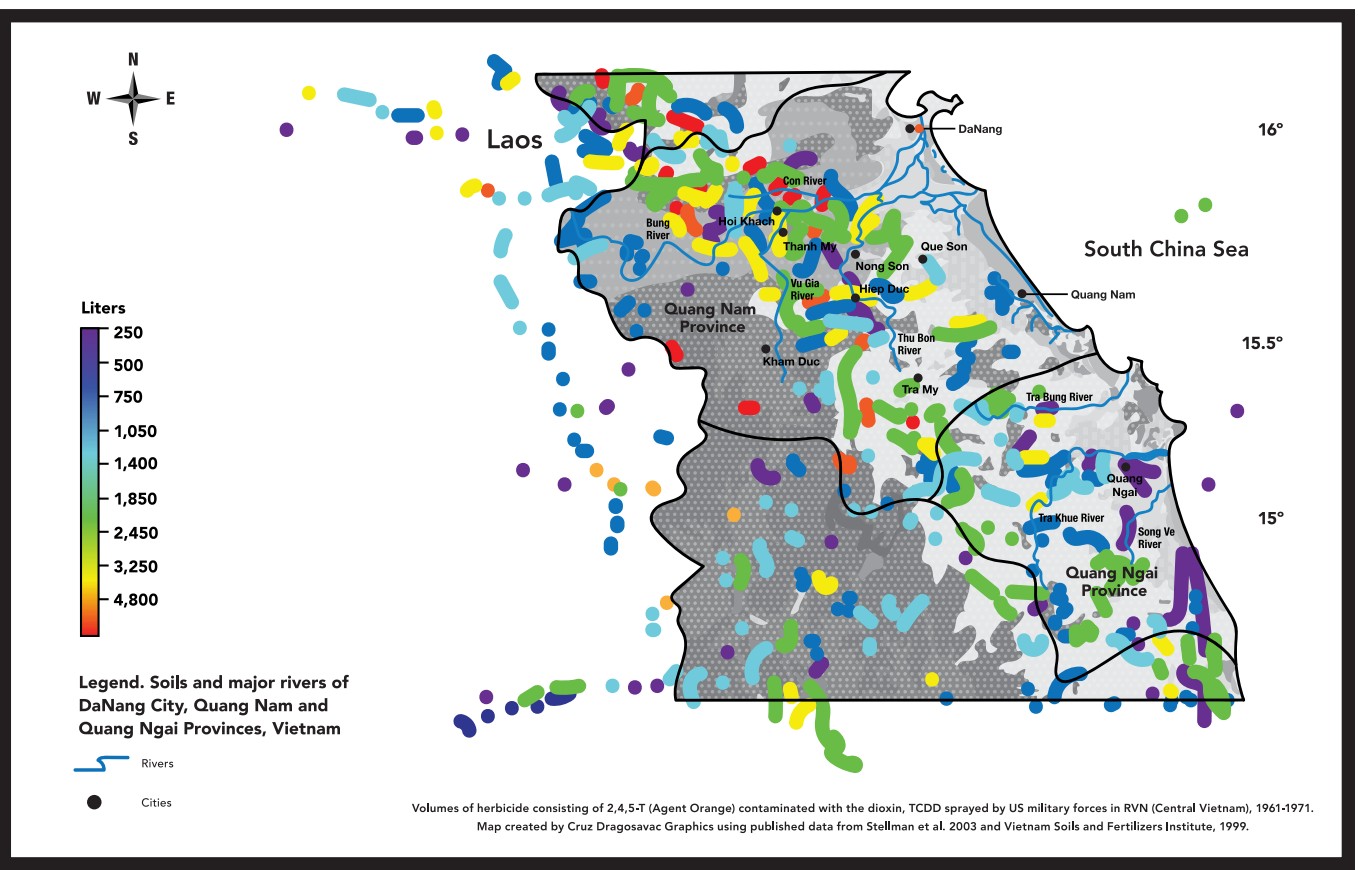

**Fig 1. Map of Da Nang City and Vietnam provinces Quang Nam and Quang Ngai in Central Vietnam represents a re-creation of geographic locations of herbicide applications in Vietnam based on Stellman et al (2003) [7] re-estimate of the volume of Agent Orange and other 2, 4, 5-T formulations sprayed between 1961 and 1971 in the Da Nang region.** The color scale to the left of the map shows increasing volume of herbicide sprayed (250 liters to more than 4,800 liters in a single location). The background shades of white and gray represent major soils groups and rivers of Da Nang, Quang Nam and Quang Ngai provinces. Lighter background shades are lowlands and upland valley locations where soils are better suited to grow agricultural crops and rural population densities are likely to be higher. Map created by Dragosavac Graphics.

wind action, runoff, soil erosion, transport and deposition, these dioxin TCDD "hotspots" have spread beyond airbase perimeter fences and into adjacent lakes, rivers and ponds [8].

Expensive remediation efforts by the United States and Vietnam have attempted to address contamination at several military base sites where the dioxin TCDD persists in the soil including an environmental remediation completed at the Da Nang Airport in November 2017 [12]. However, there is growing concern that dioxin bioaccumulation in adipose tissues of animals in aquatic and terrestrial food chains and in humans at even low levels of TCDD TEQ (toxic equivalents) from contact with contaminated soils and sediments can result in bio-magnification levels harmful to humans [13–15]. This suggests that Vietnamese who live close to the land, with daily intimate contact with its soil, sediments, and water resources may continue to be exposed to dioxin TCDD if it persists in their environment.

## Methods

### Ethics statement

The original study was approved by the Scientific and Ethics Board of DHWC and the Ethics Board of the Medical Center of Ludwig Maximilian University prior to data collection and no re-approval was required. Information on congenital anomalies was extracted from medical

records and the individual informed consent was not required by ethics committee. Research for this project complies with the World Medical Association Declaration of Helsinki regarding the ethical conduct of research involving human subjects.

## Design, data collection, data analysis

The original cross-sectional study was conducted in the neonatal unit and delivery room at DHWC in Da Nang City, the only Children's Hospital in the Da Nang region. Data were collected on 14,335 live births from April 2015 to March 2016 and included all newborns with congenital anomalies born at DHWC. No other centers in Central Vietnam contributed birth data to the original study as there is no regional or national neonatal CHD database in Vietnam and patient data sharing is not routinely done between institutions in Vietnam. At that time, there were approximately 4,000 admissions to the DHWC neonatal unit for different disease states with various congenital anomalies identified at birth making up a large proportion of those admissions. During the period of the original study, Da Nang City had a population of approximately one million people and the surrounding provinces of Quang Ngai and Quang Nam together had a population of approximately three million people. As detailed above, during the Vietnam War, these Central Vietnam regions were heavily sprayed with Agent Orange contaminated with TCDD and little to no dioxin TCDD soil testing or remediation efforts had been completed in the region when the original study data were collected.

Infant birth data including type of congenital anomaly found and parent demographics data (highest education level, parental occupations, possible herbicide exposure, etc.) were collected as described in the original study. Infants born with obvious congenital anomalies (or signs associated with a congenital anomaly) were transferred to the neonatal unit and all their data collected there. Infant data were collected in the delivery room if the infant had a minor (or non-life threatening) congenital anomaly. Any further information on all newborns identified with congenital anomalies was extracted from the medical records at the time of discharge by two research nurses and data were checked to avoid duplications between cases identified in the delivery room and the neonatal unit.

In the original study, patients suspected of having CHD underwent a complete 2D and Doppler cardiac ultrasound examination of the heart to confirm a suspected CHD diagnosis. In the current study, these original CHD data were re-reviewed in April 2021 by physicians [TTH, HN, CBC] specifically trained in pediatric cardiology and reclassified utilizing the major presenting CHD pathological lesion into the same 27 sub-types used in the 2019 Liu et al. study. Small (<4mm) ASD, hemodynamically insignificant VSD, and/or hemodynamically insignificant PDA lesions were eliminated in this reevaluation. The full Liu et al. systematic review and meta-analysis comprised 260 global studies (130,758,851 live births) grouped into North America, South America, Europe, Asia, Africa and Oceania with the Asian CHD prevalence subgroup consisting of 74 studies (52,206,662 live births). For our current study, our revised CHD birth prevalence data comparisons were made specifically to the total Asian CHD birth prevalence rate. Then, where the total number of a specific CHD subtype made the calculation reasonable, the Da Nang CHD subgroup prevalence rates were compared to the Liu et al CHD subgroup prevalence rates. Statistical analyses were performed using Python (Version 3.4) and the significance level (two-tailed) was defined at $p < 0.05$. Data are presented as prevalence rate per 1000 live births with 95% confidence intervals.

## Results

### Da Nang region CHD birth prevalence rates

**Total CHD birth prevalence rates.** The original Giang et al. study reported findings on 551 infants with all types of congenital anomalies, and of those, 288 infants were originally

found to have a CHD with a prevalence rate of 20.09/1000 live births. In our re-review of the original CHD study data as described above, we identified 211 (38.2%) infants with significant CHD utilizing newer diagnostic criteria. The current study used these revised data and found the Da Nang region total CHD birth prevalence rate to be 14.71/1000 live births (95% CI 12.74–16.69). When singular small PDAs (which are a common finding at birth and close spontaneously over the first few days of life) were eliminated, the Da Nang region CHD birth prevalence rate dropped to 12.20/1000 live births (95% CI 10.47–14.00). These data were then compared to the total CHD prevalence data for Asia published by Liu et al. of 9.34/1000 (95% CI 8.07–10.70). Statistical analysis of these two total CHD birth prevalence rates resulted in a significantly higher overall Da Nang CHD birth prevalence rate (p<0.0001) and after singular PDA were eliminated from the data (p<0.0001).

**Sub-type CHD birth prevalence rates.** Subtypes of CHD from the Da Nang region based on the revised data set are presented in Table 1. As also found in the Liu et al. study, atrial septal defects (ASD), ventricular septal defects (VSD), and patent ductus arteriosus (PDA) made up the majority of identified CHD lesions. Only the subtypes of ASD, VSD and PDAs in our current study (Table 2) had adequate numbers to calculate prevalence rates 6.55 (95% CI 5.23–7.87), 3.69 (95% CI 2.70–4.69) and 2.51 (95% CI 1.69–3.33) per 1000 live births, respectively. These CHD subtype data were also compared to the Liu et al. Asian CHD subtype prevalence rates for ASD, VSD and PDA subtypes (Table 2). While both ASD and PDA prevalence rates were significantly higher in the Da Nang region, we could not detect a significant difference in the VSD prevalence rates, but this may be a function of the lower VSD patient numbers found in our single center study as compared to the much larger VSD patient numbers in the Liu et al. Asian subgroup study.

## Discussion

It is well established that CHD is a large, rapidly emerging global problem in child health and there are significant global inequities in delivering CHD care to LMIC countries. Reported CHD birth prevalence rates from LMIC can vary significantly depending on the CHD definitions used, populations studied, what research methods are employed and whether studies

**Table 1. Distribution of CHD sub-types in the Da Nang region, 2015–2016.**

| Type of CHD (N = 211) | Number | % of total CHD |
|---|---|---|
| TOF | 1 | <1 |
| TGA (TGA/VSD) | 4 | 1.9 |
| DORV | 3 | 1.4 |
| VSD | 53 | 25.1 |
| ASD | 94 | 44.5 |
| AVC | 2 | 1 |
| PA/VSD | 2 | 1 |
| PDA | 36 | 17.1 |
| PS | 3 | 1.4 |
| Congenital MI | 1 | <1 |
| Ebstein's anomaly | 1 | <1 |
| PA/IVS | 4 | 1.9 |
| HLHS | 1 | <1 |
| Coarctation | 1 | <1 |
| IAA | 2 | 1 |
| TAPVR (all types) | 3 | 1.4 |

**Table 2. Comparison of current Da Nang region / Asian CHD Sub-type birth prevalence.**

| Lesion | Current CHD Rate Prevalence (95% CI) per 1000 births | Liu et al. CHD Rate Prevalence (95% CI) per 1000 births | p-value |
|---|---|---|---|
| All CHD | 14.71 (12.74–16.69) | 9.34 (8.07–10.70) | p<0.0001 |
| CHD (PDAs eliminated) | 12.20 (10.47–14.00) | 9.34 (8.07–10.70) | p<0.0001 |
| ASD | 6.55 (5.26–7.87) | 2.66 (1.81–3.67) | p<0.0001 |
| VSD | 3.69 (2.70–4.69) | 3.95 (3.49–4.43) | p = 0.677 |
| PDA | 2.51 (1.69–3.33) | 1.54 (1.13–2.00) | p = 0.019 |

report CHD incidence rates instead of prevalence rates which makes comparisons more difficult. Further, better CHD detection by echocardiography, especially in infants with murmurs has resulted in increased CHD diagnoses and reporting, thus changing CHD birth prevalence rates over time [16–18].

In a large systematic review and meta-analysis of 114 studies, van der Linde et al., in 2011 reported an Asian CHD birth prevalence of 9.34/1000 births (95% CI: 8.90–9.70) [19]. Studies from other areas of Southeast Asia showed a wide variation in the reported CHD birth prevalence rates. Qu et al. in 2016 examined the incidence of CHD in the Guangdong Province of southern China and reported a CHD incidence of 11.10/1000 live births [20]. Saxena et al., also in 2016, reported a CHD birth prevalence rate of 8.07/1000 live births in India. [21]. For our comparison study, we chose the Liu et al. Asian subgroup from their 2019 Global Birth prevalence of CHD defects (1970–2017) systematic review and meta-analysis for two reasons: it is the most contemporary comparison available for CHD birth prevalence rates and; we were able to specifically compare our birth data to birth data from the surrounding Asia regions making any conclusions more specific to data developed in Vietnam.

In our current study, the total CHD birth prevalence rate, as well as the subtypes of ASD and PDA prevalence rates for the Da Nang region are significantly higher than the contemporary Asian CHD birth prevalence rates as well as other CHD birth prevalence data published for Southeast Asia. While it is impossible to exactly match all the multiple factors that may affect a pregnancy and cardiac development, in comparing the contemporary Liu et al. Asian data to our Da Nang region's data, we attempted to mitigate many of the socio-economic and environmental factors common to the region. The main difference that remains between these studies are the multiple dioxin contaminated herbicides sprayed on Central Vietnam between 1961 and 1971 and the possibility of significant remaining dioxin TCDD in the environment which pregnant mothers in the Da Nang region could be exposed to. The relationship between herbicides with 2,4,5-T and the contaminate dioxin TCDD sprayed over South Vietnam including high concentrations in the Da Nang region and CHD have not been substantially investigated previously. A joint US-Vietnam study was proposed in 2003 to assess the association of birth defects in children whose mothers were exposed to Agent Orange but was never undertaken as the funding deadline expired before the study could take place [22].

With recent advances in genetic research, approximately one-third of all CHD cases have been shown to have a simple genetic cause. The remaining cases can also be caused by oligogenic factors, environmental factors, and/or gene-environmental interactions and can be broadly divided into two categories: extrinsic factors, such as teratogen exposure and nutrient deficiencies, and intrinsic factors, including maternal disease and illness [23]. Normal embryonic cardiac development is a complex process that is vulnerable to perturbation by external factors and the interruption of that process can result in a variety of cardiac structural anomalies depending on the insult and timing. Understanding of the embryonic mechanisms by

which dioxins exert their cardiotoxic effects in humans is limited, and a direct causal link between congenital heart disease and fetal exposure to dioxins has yet to be established [24].

However, there does exist in the literature possible associations of dioxin TCDD on CHD. In a study of infants born to mothers living near incinerators emitting complex mixtures of dioxins, furans, particulates, and heavy metals, a higher incidence of cardiac malformations and lethal congenital heart disease were found [25]. In other studies, the incidence of hypoplastic left heart syndrome was epidemiologically associated with maternal exposure to halogenated hydrocarbons, dioxins, and polychlorinated biphenyls during pregnancy [26,27]. Loffredo, in a 2001 study, suggests that risk of transposition of the great arteries (TGA), a unique cardiovascular malformation, is increased with maternal exposure to herbicides and rodenticides during the periconceptional period of pregnancy [28]. Further, Puga states that there is ample evidence that dioxin TCDD causes persistent cardiac defects in zebrafish, chickens, mice, and likely humans and is associated with human cardiovascular disease. Interestingly, in our study as in previous reports, ASD is the most common defect seen; however, there is no environmental toxin that has been reported to be specifically associated with any CHD lesion [29].

Environmental persistence and human chronic exposure to the dioxin TCDD are two factors that may explain some of South Vietnam's current high prevalence rates of disease and disabilities. Persistent dioxin TCDD concentrations in the Vietnam environment are a function of past herbicide spray frequency and distribution, as well as conditions that can slow or prevent decomposition, re-distribution and recycling within the ecosystem, and bioavailability [11]. Further, dioxin TCDD half-life in soil more than an inch below the surface has been estimated at 20 to 100 years depending on geographic conditions [9]. Soil samples collected in 2017 revealed dioxin TCDD residuals remain in topsoil samples from A-So air base north of Da Nang where Agent Orange was stored and used during the Vietnam War [30].

The half-life of dioxin TCDD and its persistence in the environment also depends on where it is deposited, the characteristics of the soil and sediments to which it adheres, microbial action and exposure to sunlight. Dioxin TCDD degrades when fully exposed to sunlight; degradation is slowed considerably when dioxin TCDD contaminated soils and sediments are buried or sheltered from direct sunlight. Dioxin TCDD molecules are not absorbed into soil particles but rather they adhere to the surfaces of soil particles [7,8,11,15,31]. When dioxin TCDD attaches to colloidal organic or humic soil materials, it can be mobilized to a new location during the monsoon season and resulting periods of high-water run-off. Herbicide-defoliated fields, forests, and stream banks "naturally" erode barren soils and transport them to the lowest spots in the landscape [32]. Soils in these low spots can become a sink for dioxin TCDD, as this contaminate is hydrophobic and dioxin TCDD does not dissolve in water but remains attached to soil and sediment particles. Further, when sediments are re-suspended in water during flooding, wave action, wind erosion, and farming activities and redeposited, residual dioxin TCDD may seldom be exposed to sunlight.

Agriculture, forestry, and fishing provide many Vietnamese households with subsistence food security and livelihoods; and often are pathways out of poverty [33]. Rice and fish are primary food sources for Vietnamese households and many small-scale farmers continue traditional "soil puddling" practices in their rice paddy fields [34]. Rice agriculture in Vietnam is a hands-on (and feet) occupation with constant soil contact. A Dwernychuk et al. study reaffirmed the apparent food chain transfer of dioxin TCDD from contaminated soils to fish and duck tissues, then to humans as measured in the whole blood and breast milk [11]. Any contamination in paddy soils or fish-pond sediments could easily bring humans into contact via skin and/or hands with potential ingestion through farming, food preparation and consumption activities [15].

Hot-spot and superfund cleanup criteria, methodologies and dioxin risk assessments in the United States and other developed countries have focused on cleanup of residential and industrial sites and likely exposure scenarios [35] that may not apply to the entire Da Nang region. Paustenbach et al. cautions that dioxin mitigation work in urban residential settings of developed countries and assumptions about soil exposure and pathways to human impact models does not apply to subsistence agriculture, fisheries, grazing, and livestock raising practices in which farmers and their families who may have chronic and direct skin contact with dioxin contaminated soils and by indirect pathways into the food chain [11,31].

Finally, Vietnam is a country with a significant population of females of childbearing age. Females between the ages of 19 and 39 in 2015 made up approximately 34.2% of the female population with birth rates that varied between 34.9 births/1000 females (35–39 years old) and 135.1 births/1000 females (25–29 years old) [36,37]. Agriculture in rural areas is a primary occupation for both men and women, with 63% of working women and 57% of working men engaged in agriculture, forestry, and fisheries. The traditional job option for women is agriculture and hand labor in agriculture dominates Vietnamese production systems [38]. Thus, there is a very high potential risk of exposure of childbearing age females to contaminated soil and sediments in areas where dioxin TCDD persists in the environment. The exposure pathways and health impacts to farmers and rural communities whose occupations and cultural practices entail daily contact over a lifetime with dioxin TCDD in the soil and sediments is a knowledge gap that has not been fully explored and is not well understood [15].

This current study has several limitations. Most importantly, the calculated CHD prevalence rate in our study comes from live births presenting at a single center (DHWC) as CHD birth data-sharing is not routinely done in Vietnam, so may underestimate the true CHD birth prevalence due to uncounted CHD deliveries at other health care facilities in Central Vietnam. Secondly, our data only reflect those infants in whom CHD was suspected/detected at the time of birth and could have missed infants who were diagnosed after discharge. There is also a bias in our comparison of the Da Nang area CHD birth prevalence data to the Liu et al. CHD birth prevalence data as their reported CHD "birth" prevalence included studies of children identified with CHD from ages 0–6 years old. Finally, we can only suggest an association between dioxin TCDD and CHD, not a direct correlation. While expensive serum dioxin tests do exist, the 'association' we suggest between dioxin TCDD and early fetal cardiac development would probably not yield a positive result if every CHD infant at birth was tested; however, it is not economically feasible to test every pregnant female in the Da Nang region.

## Conclusions

This study highlights the elevated CHD birth prevalence rate found in the Da Nang region of Central Vietnam and raises an important question as to a possible environmental cause not found in surrounding LMIC Asian countries. While difficult to account for the many factors which might influence embryonic cardiac development, the one factor which stands out in this CHD prevalence birth rate comparison is the possible on-going exposure of pregnant mothers to dioxin TCDD from the soils and sediments in the Da Nang region, and bioaccumulation in the food chain and their diets with potential to expose their fetus to the toxic effects of TCDD. To date, a one-to-one cause and effect has not been proven; however, our findings suggest an urgent need for research that closely examines the association between residual dioxin TCDD in the environment and elevated CHD birth prevalence in the Da Nang region. Mitigation strategies for dioxin TCDD were completed in Da Nang City at the end of 2017, but there remains the potential for significant dioxin TCDD exposure in the surrounding provinces. Assessment of TCDD levels in soil/sediments in high volume spray rural locations,

exposure mechanisms, and evaluation of toxic dose levels (TEQs) that have potential to affect CHD birth prevalence rates are critical to increase our understanding of TCDD health risks to infants and children. Finally, given that there is a known elevated CHD birth prevalence rate in this region, a "new" public health strategy might be to consider enhanced fetal cardiac echocardiography to screen pregnancies for CHD and especially complex CHD (e.g., d-TGA) requiring early interventions for optimal outcomes.

## Acknowledgments

We would like to thank Dr. Bernard D. Keavney, Professor of Cardiovascular Medicine in the Division of Cardiovascular Sciences at the University of Manchester, UK for sharing his Asian CHD sub-group prevalence data. We also would like to thank Dr. Tran Thi Hoang and Ms. Dang Thi My Na at Da Nang Hospital for Women and Children for their support during the time of data collection.

## Author Contributions

**Data curation:** Hoang Thi Nam Giang, Tran Thanh Hai, Hoang Nguyen, Casey B. Culbertson.

**Formal analysis:** Hoang Nguyen.

**Investigation:** Hoang Thi Nam Giang.

**Methodology:** Hoang Thi Nam Giang, Casey B. Culbertson.

**Validation:** Tran Thanh Hai, Thanh K. Vuong, Lois Wright Morton.

**Writing – original draft:** Hoang Thi Nam Giang, Lois Wright Morton, Casey B. Culbertson.

**Writing – review & editing:** Hoang Thi Nam Giang, Tran Thanh Hai, Hoang Nguyen, Thanh K. Vuong, Lois Wright Morton, Casey B. Culbertson.

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
