## [Decision Letter · Decision Letter 0]

28 Jul 2022

PGPH-D-22-00454

Elevated congenital heart disease birth prevalence rates found in Central Vietnam and dioxin TCDD residuals from the use of 2, 4, 5-T herbicides (Agent Orange) in the Da Nang region.

Dear Dr. Hoang Giang,

Thank you for submitting your manuscript to PLOS Global Public Health. After careful consideration, we feel that it has merit but does not fully meet PLOS Global Public Health’s publication criteria as it currently stands. Therefore, we invite you to submit a revised version of the manuscript that addresses the points raised during the review process.

A rebuttal letter that responds to each point raised by reviewer(s). You should upload this letter as a separate file labeled 'Response to Reviewers'.A marked-up copy of your manuscript that highlights changes made to the original version. You should upload this as a separate file labeled 'Revised Manuscript with Track Changes'.An unmarked version of your revised paper without tracked changes. You should upload this as a separate file labeled 'Manuscript'.

We look forward to receiving your revised manuscript.

Kind regards,

Jose Ignacio Nazif-Munoz, Ph.D.

Academic Editor

Journal Requirements:

1. In the online submission form, you indicated that "All data will be available on request". All PLOS journals now require all data underlying the findings described in their manuscript to be freely available to other researchers, either 1. In a public repository, 2. Within the manuscript itself, or 3. Uploaded as supplementary information.

2. Figure 1 appears to have been adaptedfrom a previously published figure. Please provide written permission from the copyright holder to publish this under our CC-BY 4.0 license, or remove the figure / replace the image. Please note we do not recommend using standard request forms available on Publishers' websites, as they grant single use rather than republication under an open access license.

Reviewers' comments:

Reviewer's Responses to Questions

**Comments to the Author**

1. Does this manuscript meet PLOS Global Public Health’s publication criteria? Is the manuscript technically sound, and do the data support the conclusions? The manuscript must describe methodologically and ethically rigorous research with conclusions that are appropriately drawn based on the data presented.

Reviewer #1: Yes

Reviewer #2: Partly

2. Has the statistical analysis been performed appropriately and rigorously?

Reviewer #1: Yes

Reviewer #2: Yes

3. Have the authors made all data underlying the findings in their manuscript fully available (please refer to the Data Availability Statement at the start of the manuscript PDF file)?

Reviewer #1: Yes

Reviewer #2: No

4. Is the manuscript presented in an intelligible fashion and written in standard English?

Reviewer #1: Yes

Reviewer #2: No

5. Review Comments to the Author

Reviewer #1: The manuscript is informative and merits publication to bring about awareness for prevention of CHD caused by conscious and unconscious presence of dioxin TCDD residuals from the use of 2, 4, 5-T herbicides (Agent Orange)in the soil.

However it is expected that the author should provide full forms of Types of CHD in Table 1. Distribution of CHD sub-types in the Da Nang Region, 2015-201

Reviewer #2: The study is important in the context of conflict-prone region. However, the manuscript should be improved substantially before accepting for publication.

Please follow the journal instruction to write the Method section.

The methodological description are not comprehensive.

Line 148: authors repeatedly mentioned that in the "original study" it is not clear how this study is different the previous one.

I would suggest to include if authors tested statistical difference in the sub-group analysis.

Please present effect estimates in two digits prior to and after decimal point

The manuscript is poorly written or interpreted the results. Authors were more interested in highlighting the p-values rather than interpretating the results. The comparison authors made with previous study should be clear and concise because readability is low. For example in line 179-186: "This resulted in a revised Da Nang region overall CHD birth prevalence rate 14.712/1000 live births 177 (95% CI 12.747-16.690) and when eliminating singular PDAs, which are a common finding at

birth, the Da Nang region CHD birth prevalence rate dropped to 12.207/1000 live births (95% CI 10.470-14.005). These data were then compared to CHD prevalence data for Asia published by Liu et al. of 9.342/1000 (95% CI 8.072-10.704). Statistical analysis of these two CHD birth prevalence rates resulted in a significantly higher overall Da Nang CHD birth prevalence rate

(p<0.0001) and when singular PDA were eliminated (p<0.0001)."

Findings will be more readable if authors could revise it by a language expert. I apologize if it sounds rude.

Discussion should be precise because most of the information are repeated.

6. PLOS authors have the option to publish the peer review history of their article (what does this mean?). If published, this will include your full peer review and any attached files.

**Do you want your identity to be public for this peer review?** For information about this choice, including consent withdrawal, please see our Privacy Policy.

Reviewer #1: **Yes: **Prof Anita Saxena

Reviewer #2: No

---

## [Editor Report · Decision Letter 1]

19 Sep 2022

Elevated congenital heart disease birth prevalence rates found in Central Vietnam and dioxin TCDD residuals from the use of 2, 4, 5-T herbicides (Agent Orange) in the Da Nang region.

PGPH-D-22-00454R1

Dear Dr Giang,

We are pleased to inform you that your manuscript 'Elevated congenital heart disease birth prevalence rates found in Central Vietnam and dioxin TCDD residuals from the use of 2, 4, 5-T herbicides (Agent Orange) in the Da Nang region.' has been provisionally accepted for publication in PLOS Global Public Health.

Best regards,

Jose Ignacio Nazif-Munoz, Ph.D.

Academic Editor
